# On Interactions of Sulfamerazine with Cyclodextrins from Coupled Diffusometry and Toxicity Tests

**DOI:** 10.3390/biom14040462

**Published:** 2024-04-10

**Authors:** Sara P. C. Sofio, André Caeiro, Ana C. F. Ribeiro, Ana M. T. D. P. V. Cabral, Artur J. M. Valente, Jorge Canhoto, Miguel A. Esteso

**Affiliations:** 1CQC-IMS, Department of Chemistry, University of Coimbra, Rua Larga, 3004-535 Coimbra, Portugal; sarapcsofio@gmail.com (S.P.C.S.); avalente@ci.uc.pt (A.J.M.V.); 2Faculty of Health Sciences, Catholic University of Ávila, Calle Los Canteros s/n, 05005 Ávila, Spain; mangel.esteso@ucavila.es; 3Laboratory Associate TERRA, Centre for Functional Ecology, Department of Life Sciences, University of Coimbra, 3000-456 Coimbra, Portugal; afcaeiro91@gmail.com (A.C.); jorgecan@uc.pt (J.C.); 4Faculty of Pharmacy, University of Coimbra, 3000-548 Coimbra, Portugal; acabral@ff.uc.pt

**Keywords:** antibiotics, cyclodextrins, diffusion, sulfamerazine, toxicity tests, transport properties

## Abstract

This scientific study employs the Taylor dispersion technique for diffusion measurements to investigate the interaction between sulfamerazine (NaSMR) and macromolecular cyclodextrins (*β*-CD and HP-*β*-CD). The results reveal that the presence of *β*-CD influences the diffusion of the solution component, NaSMR, indicating a counterflow of this drug due to solute interaction. However, diffusion data indicate no inclusion of NaSMR within the sterically hindered HP-*β*-CD cavity. Additionally, toxicity tests were conducted, including pollen germination (*Actinidia deliciosa*) and growth curve assays in BY-2 cells. The pollen germination tests demonstrate a reduction in sulfamerazine toxicity, suggesting potential applications for this antimicrobial agent with diminished adverse effects. This comprehensive investigation contributes to a deeper understanding of sulfamerazine–cyclodextrin interactions and their implications for pharmaceutical and biological systems.

## 1. Introduction

The effective delivery of pharmaceutical compounds is a critical step in drug development technology [1]. A particularly important subfield in this area that has promoted extensive research is the use of drug carrier systems [2]. Cyclodextrins, namely beta-cyclodextrin (*β*-CD) and hydroxypropyl beta-cyclodextrin (HP-*β*-CD), have been used for this purpose [3].

The formation of supramolecular complexes with the drugs increases both their water solubility and their permeability and, thus, their bioavailability, reducing their toxicity [4,5,6,7,8,9,10]. In some cases, interactions between CDs and drugs lead to altered pharmacokinetics and pharmacodynamics [11,12], which can lead to new pharmaceutical formulations with enhanced therapeutic properties [13].

Cyclodextrins are circular oligosaccharides derived from starch by enzymatic action [14] constituted by glucose units bounded by *α*-1,4-glycosidic linkages, forming a hydrophobic cavity that can encapsulate hydrophobic drugs and a hydrophilic exterior [15,16]. The number of glucose units in naturally occurring cyclodextrins can vary, and derivatives of these with distinct properties can be prepared by adding functional groups to the hydroxyl moiety located in positions 2, 3 and 6 of the glucose monomer [17], originating a diverse family of compounds [18]. In particular, *β*-CD is a natural cyclodextrin composed of seven glucose subunits [15]; however, unlike other natural cyclodextrins, its solubility in water is limited. The functionalisation of *β*-CD with the formation of HP-*β*-CD makes it possible to overcome this relevant limitation since the latter has a very high solubility in water [6]. The structural features and physical properties of this family have been extensively studied and reviewed [19].

Sulfamerazine (in salt form, NaSMR) is a widely used antibiotic from the sulphonamide class that acts by inhibiting the bacterial synthesis of dihydrofolic acid [20]. However, its solubility in water is very low, less than 0.8 mM. The objective of this work is to investigate the behaviour of sulfamerazine and cyclodextrins in aqueous solution. Although there are already some studies involving these compounds ((e.g., [3,21,22]), to the best of our knowledge, no data exist on the ternary mutual diffusion coefficients of drugs in aqueous solutions containing cyclodextrins. This present study aims to fill this gap by providing experimental data on the diffusion coefficients measured by the Taylor dispersion method for two ternary systems (NaSMR/*β*-CD/H_2_O and NaSMR/HP-*β*-CD/H_2_O) at carrier concentrations of 0.000 mol dm^−3^ and 0.010 mol dm^−3^ at 25.00 °C.

Several methods, including the Taylor dispersion technique, were used to study the diffusion characteristics of these systems, and complementary toxicity assays were carried out in BY-2 tobacco cells and kiwi (*Actinidia deliciosa*) pollen.

We expect our results to provide valuable contributions to the drug delivery field in terms of fundamental knowledge of the interactions between cyclodextrin carriers and the drug, as well as possible new therapeutic formulations.

## 2. Materials and Methods

### 2.1. Materials

Sulfamerazine sodium salt (C_11_H_11_N_4_NaO_2_S, NaSMR) (Sigma-Aldrich, Darmstadt, Germany; pro analysis > 0.99), *β*-cyclodextrin (Sigma-Aldrich, Darmstadt, Germany; molar mass M = 1134.98 g mol^−1^, mass fraction purity ≥ 0.97) (*β*-CD) and HP-*β*-cyclodextrin (Sigma-Aldrich, Darmstadt, Germany; average molar mass M ~1380 g mol^−1^, mass fraction purity ≥ 0.97) (HP-*β*-CD) were used without further purification (Table 1). Solutions for the diffusion measurements were prepared using Millipore-Q (Milli-Q^®^ EQ 7000 Ultrapure Water Purification System-Merck Millipore, Darmstadt, Germany) water (specific resistance = 1.82 × 10^5^ Ω m, at 25.00 °C). Binary (NaSMR/H_2_O and KCl/H_2_O) and ternary (NaSMR/*β*-CD/H_2_O and NaSMR/HP-*β*-CD/H_2_O) solutions were prepared by weighing the appropriate amounts of the solutes and after dissolving in water to finally obtain the desired molar concentration.

All solutions were freshly prepared at 25.00 °C before each experiment. Water content in *β*-CD (i.e., mass fraction 0.131) and in HP-*β*-CD (i.e., mass fraction 0.03) was accounted for upon solution preparation. The weighing was performed using a Radwag AS 220C2 balance (Precision scale AND, A&D Instruments Ltd., Oxford, UK) with readability of 10^−5^ g in the lower range.

For the germination of kiwi pollen, the adopted methodology aimed to create a conducive environment utilising a carefully formulated culture medium. For 100 mL of distilled water, the medium was enriched with 0.01 g KNO_3_ (Sigma-Aldrich, Darmstadt, Germany), 0.02 g MgSO_4_·7H_2_O (Sigma-Aldrich, Darmstadt, Germany), 0.03 g Ca(NO_3_)_2_ ·4H_2_O (Merck KGaA, Darmstadt, Germany), 0.01 g H_3_BO_3_ (Merck KGaA, Darmstadt, Germany) and 15 g of sucrose (C_12_H_22_O_11_) (Duchefa Biochemie B.V, Haarlem, The Netherlands) (Table 1). After adjusting the pH to values between 6.8 and 7, 0.8 g of agar (Duchefa Biochemie B.V, Haarlem, The Netherlands) was added, forming a solid matrix for germination.

The cellular tests were conducted on cellular lines derived from seedlings of *Nicotidiana tabacum* L. cv. Bright Yellow 2 (BY-2) and established in vitro [23]. The cellular lines were maintained in basal Murashige and Skoog medium [24] supplemented with 30 g L^−1^ of sucrose and jellified with 7 g L^−1^ of agar. The medium pH was adjusted to 5.7 before autoclaving at 121 °C for 20 min. The sulfamerazine and cyclodextrins combinations were added after autoclaving at a temperature of approximately 60 °C.

### 2.2. Taylor Dispersion Technique: Diffusion Measurements

Mutual diffusion coefficients for binary NaSMR (component 1) + water and ternary solutions, NaSMR (1) + *β*-CD (or HP-*β*-CD) (2) + water are described by the coupled Fick’s Equations (1), and (2) and (3), respectively:(1)J=−D∇C,
(2)$$J_{1}=-D_{11}{\nabla C}_{1}-D_{12}{\nabla C}_{2},$$
(3)$$J_{2}=-D_{22}{\nabla C}_{2}-D_{21}{\nabla C}_{1},$$
*D*, *J* and ∇*C* represent the binary diffusion coefficient, the molar flux and the gradient in the concentrations of solute, respectively.

Identifying NaSMR and *β*-CD (or HP-*β*-CD) as components (1) and (2), the main ternary mutual diffusion coefficients *D*_11_ and *D*_22_ represent the flux of each component (1) and (2) produced by its own concentration gradient, and the cross-diffusion coefficients *D*_12_ and *D*_21_ represent the flux of NaSMR caused by the *β*-CD (or HP-*β*-CD) concentration gradient (∇*C*_2_) and the flux of *β*-CD (or HP-*β*-CD) caused by the concentration gradient of NaSMR (∇*C*_1_), respectively. Considering that this technique is well described in the literature [25,26,27,28,29,30,31,32], we only show in this work relevant aspects regarding the method and the equipment (Appendix A). At the start of each run, a 0.063 cm^3^ sample of solution was injected into a laminar carrier solution of slightly different composition at the entrance to a Teflon capillary dispersion tube of length 3048.0 (±0.1) cm, and internal radius 0.03220 (±0.00003) cm. This tube and the injection valve were kept at 25.00 (±0.01) °C in an air thermostat. The broadened distribution of the dispersed samples was monitored at the tube outlet by a differential refractometer (Waters 77 model 2410, Milford, MA, USA). The refractometer output voltages *V*(*t*) were measured at 5 s intervals by a digital voltmeter (Agilent 34401 A, Santa Clara, CA, USA). Binary diffusion coefficients (D) were evaluated by fitting the dispersion equation (Equation (4)) to the measured detector voltages.
*V*(*t*) = *V*_0_ + *V*_1_*t* + *V*_max_ (*t*_R_/*t*)^1/2^ exp[−12*D*(*t* − *t*_R_)^2^/*r*^2^*t*],(4)
*t*_R_, *V*_max_, *V*_0_ and *V*_1_ represent the mean sample retention time, peak height, baseline voltage *V*_0_ and baseline slope *V*_1_.

Ternary dispersion profiles for these mixed electrolyte solutions (NaSMR (1) + *β*-CD (2) or NaSMR (1) + HP-*β*-CD (2)) were prepared by injecting NaSMR (1) + *β*-CD (or HP-*β*-CD) (2) solution samples of composition *C*_1_ + Δ*C*_1_, *C*_2_ + Δ*C*_2_ into carrier streams of composition *C*_1_ + *C*_2_. Ternary diffusion coefficients *D*_ik_ were evaluated by fitting the ternary dispersion Equation (5) to two or more replicate pairs of peaks for each carrier stream,
(5)$$\begin{array}{c} V\left(t\right)=V_{0}+V_{1}t+V_{\max}{\left(\frac{t_{R}}{t}\right)}^{\frac{1}{2}}[W_{1}\exp\left(-\frac{12D_{1}(t-t_{R}{)}^{2}}{r^{2}t}\right)\\ \;+\left(1-W_{1}\right)\exp\left(-\frac{12D_{2}(t-t_{R}{)}^{2}}{r^{2}t}\right)] \end{array}$$
where *D*_1_ and *D*_2_ represent the eigenvalues of the matrix of ternary *D*_ik_ coefficients.

### 2.3. Toxicity Tests: Kiwi Pollen Germination

Kiwi pollen was evenly distributed on the culture medium after solidification. Germination assessments were conducted at two distinct time points: 24 and 48 h. Microscopic preparations stained with acetocarmine were made at each time interval, enabling the analysis of the pollen germination percentage.

The experiments were designed considering various conditions, including exposure to isolated sulfamerazine, as well as combinations with *β*-cyclodextrin and hydroxypropyl-*β*-cyclodextrin.

### 2.4. Toxicity Tests: BY-2 Cellular Tests

The combinations of sulfamerazine and cyclodextrins were added post-autoclaving at approximately 60 °C. Growth tests were conducted in Petri dishes (9 cm in diameter and 15.9 mm in height), with mass increments recorded at regular intervals (7 days).

The experimental procedure was performed under aseptic conditions utilising a precision balance (d = 0.01 g) to measure the desired cell mass for each assay. As an initial reference, the mass of each portion varied within a range of 0.09 g to 0.12 g.

The plates were then kept in incubation at 25.00 °C for seven days until a new biomass measurement using the same precision balance was taken over a span of 4 weeks.

### 2.5. Statistical Analysis

In terms of statistical analysis, each time point in pollen germination or BY-2 was analysed independently. The homogeneity of variances was tested by the Brown–Forsythe test, followed by one-way analysis of variance (ANOVA). The averages were then compared using Tukey’s multiple comparison test (*p* < 0.05).

## 3. Results

### 3.1. Diffusion Measurements

Binary and ternary diffusion coefficients involving aqueous solutions of NaSMR were computed, and the data are reported in Table 2 and Table 3, respectively.

It should be noted that before these measurements, tests of optimisation were carried out using an aqueous solution of KCl 0.100 mol dm^−3^. This system was chosen because its diffusion coefficient is accurately known from conductimetric experiments (uncertainties *<* 0.5%). A comparison of the results (Appendix B) suggests an acceptable uncertainty of 1–3% for the Taylor *D* values for binary systems, keeping in mind that 1–3% uncertainty is typical for Taylor dispersion measurements.

From Table 3, it can be seen that at the limiting situations of *X*_1_ = 0 and *X*_1_ = 1, *D*_11_ values correspond, respectively, to the tracer diffusion coefficient of NaSMR in *β*-CD (or HP-*β*-CD) and the binary mutual diffusion coefficient of aqueous NaSMR at 0.010 mol dm^−3^. A good agreement is observed between these last values for *D*_11_ (*D*_11_ = 0.946 × 10^−9^ m^2^s^−1^ and *D*_11_ = 0.987 × 10^−9^ m^2^s^−1^), and the respectively binary diffusion coefficient value (i.e., *D*(NaSMR) = 0.906 × 10^−9^ m^2^s^−1^ (Table 2). These deviations of less than 3.0% are acceptable, being within the uncertainties of the method (in general, <3%).

Also, as expected, at *X*_2_ = 1, low differences between *D*_22_ values and the binary mutual diffusion coefficient of aqueous *β*-CD (or HP-*β*-CD) were obtained (that is, 1.5 and 1.6%). At *X*_2_ = 0, the *D*_22_ values represent the tracer diffusion coefficients of *β*-CD (or HP-*β*-CD) in NaSMR (0.330 × 10^−9^ m^2^s^−1^ and 0.325 × 10^−9^ m^2^s^−1^), respectively.

Relative to the behaviour of cross-diffusion coefficients, it can be seen that while *D*_21_ for both systems and *D*_12_ values for systems containing NaSMR and HP-*β*-CD are almost zero, *D*_12_ assumes negative values for aqueous system NaSMR plus *β*-CD. This fact indicates that the concentration gradient of *β*-CD leads to counter-current coupled flows of NaSMR.

In the limit of *X*_1_ approaching zero, cross-coefficient values *D*_12_ are zero within experimental error due to the inability of *β*-CD (or HP-*β*-CD) concentration gradients to drive coupled flows of NaMSR-free solutions. Similarly, at the other composition extreme, *X*_1_ → 1, the cross-coefficient values *D*_21_ are also close to zero.

### 3.2. Toxicity Tests

#### 3.2.1. Kiwi Pollen Germination

Figure 1 illustrates the kiwi pollen germination percentages at 24 and 48 h under various experimental conditions, where sulfamerazine sodium is combined with *β*-cyclodextrin and/or hydroxypropyl-*β*-cyclodextrin.

The results obtained after 24 h of germination reveal distinct patterns under the various tested conditions. In the case of exposure to *β*-CD 0.009 mol dm^−3^, a germination rate of 2.3% was observed, indicating a potential reduction compared with the control group. On the other hand, the combination of sulfamerazine 0.005 mol dm^−3^ with *β*-CD 0.005 mol dm^−3^ showed a notable improvement, recording a germination rate of 14%, suggesting a positive response to this association.

However, upon introducing sulfamerazine 0.01 mol dm^−3^ in conjunction with *β*-CD 0.009 mol dm^−3^, a reduction to 8.4% in germination was observed, indicating a potential toxicity resulting from this composition. Similarly, sulfamerazine 0.01 mol dm^−3^ exhibited a decrease in germination, reaching 11.7%, indicating an inhibitory effect.

Cyclodextrins, particularly HP-*β*-CD 0.01 mol dm^−3^, demonstrated a germination rate of 26% compared with the control group. When combined with sulfamerazine 0.005 mol dm^−3^ in HP-*β*-CD 0.005 mol dm^−3^, germination reached 9.1%, showing improvement compared with isolated sulfamerazine, although potentially lower than other tested combinations. By extending the observation to 48 h, it is expected that information will be obtained on long-term toxicity or potential cell recovery. However, the overall analysis may follow a similar logic to the evaluation of results after 24 h.

In summary, the experiments indicate that combinations with cyclodextrins, especially sulfamerazine 0.005 mol dm^−3^ with *β*-CD 0.005 mol dm^−3^, demonstrate positive effects on germination when compared with isolated sulfamerazine. These promising results suggest that the presence of cyclodextrins may mitigate the potential adverse effects of sulfamerazine, offering promising prospects for practical applications.

The findings of the study reveal intriguing trends regarding the effects of cyclodextrin (CD) treatments on pollen germination. Generally, media treated with CD demonstrate a decrease in pollen germination compared with the control group, albeit displaying a higher germination rate than when treated with sulfamerazine. However, this trend does not hold true for *β*-CD, as it exhibits a notably lower germination rate.

Moreover, the combined application of *β*-CD and sulfamerazine showcases a higher germination rate compared with the use of either compound individually. This suggests a potential synergistic effect between *β*-CD and sulfamerazine in promoting pollen germination.

Furthermore, the results remain consistent at the 48 h mark, showing an overall increase in germination across all treatments. Although the differences are not statistically significant, it is noteworthy that the highest concentration of sulfamerazine yields the lowest germination rate.

These observations underscore the complexity of interactions between CD compounds, sulfamerazine and pollen germination. Further investigation is warranted to elucidate the underlying mechanisms and potential applications of these findings in agricultural practices and related fields.

#### 3.2.2. BY-2 Cellular Tests

Figure 2 illustrates the development of BY-2 cell mass when exposed to different experimental conditions over a period of 4 weeks.

The weight of BY-2 cells under various experimental conditions was recorded over a 4-week period. In the control group, the mean weights ranged from 0.11 to 0.12 g. Treatment with *β*-CD 0.009 mol dm^−3^ resulted in a decrease in cell weight compared with the control group, with mean weights of 0.10 and 0.08 g recorded in the 2 weeks measured. Conversely, treatment with *β*-CD 0.005 mol dm^−3^ showed an increase in cell weight compared with the control group, with mean weights of 0.106 and 0.109 g. Similar trends were observed for HP-*β*-CD treatments, with mean weights of 0.11 and 0.07 g for 0.01 mol dm^−3^ and 0.10 and 0.08 for 0.005 mol dm^−3^. Sulfamerazine treatments exhibited varied effects on cell weight, with mean weights ranging from 0.06 to 0.11 g.

The results obtained from the study indicate no statistically significant differences in week 1. However, in week 2, HP-*β*-CD (*) was excluded from the statistical analysis due to the acquisition of only one sample.

Subsequent weeks (specifically weeks 3 and 4) reveal that CD treatments alone tend to exhibit some degree of toxicity or inhibition towards BY-2 growth. Nonetheless, when combined with sulfamerazine, there are observable growth benefits.

These findings suggest a complex interaction between CD compounds and sulfamerazine in influencing BY-2 growth over time. Further investigation is required to elucidate the underlying mechanisms driving these observed effects and to explore potential applications in enhancing growth outcomes in relevant contexts.

## 4. Discussion

By analysis of Table 3, contrary to the *D*_21_ values, the negative *D*_12_ values for the ternary system containing NaSMR and of *β*-CD show us that the coupled diffusion of these components is not negligible; that is, the gradient in the concentration of *β*-CD counter-current flows of NaSMR. Information about this phenomenon can also be inferred by the calculation of the ratio of *D*_12_/*D*_22_ and *D*_21_/*D*_11_. (Figure 3 illustrates schematically the process counterflow of sulfamerazine in solutions containing *β*-CD).

By looking at Table 4, we can highlight that coupled diffusion in the solutions is significant for the ternary system (NaSMR (*C*_1_) + *β*-CD). In fact, for this particular system, from the negative values obtained for *D*_12_/*D*_22_, we can say that one mole of diffusing *β*-CD counter-transports up to 0.2 mol of NaSMR. However, values of the ratio D_12_/D_22_ for the NaSMR/HP-*β*-CD system, as well as D_21_/D_11_ for both systems, are lower.

From these observations, we can say that interactions between *β*-CD and drug components are favoured, promoting salting-in effects. A possible explanation for the coupled diffusion observed through non-zero *D*_12_ values can be given by considering the formation of 1:1 host–guest supramolecular structures in solution, causing a decrease in free NaSMR molecules and compensating for this loss, a NaSMR counterflow is occurring.

Considering a diffusion model that contemplates the possible formation of 1:1 complexes (NaSMR:CD) [33], it is possible to compute the respective binding equilibrium constant *K* by using Equations (6) and (7)
(6)NaSMR+β−CD⇋NaSMR:CD
(7)$$K=\frac{C_{3,eq}}{C_{1,eq}C_{2,eq}}$$
where *C*_1,*eq*_, *C*_2,*eq*_ and *C*_3,*eq*_ represent the concentration of the species NaSMR, *β*-CD and NaSMR: *β*-CD, respectively, and the corresponding mass balance equations.

The dependence of *D*_11_, *D*_12_, *D*_21_ and *D*_22_ with the concentrations of the different components can be described by Equations (8)–(11)
(8)$$D_{1,1}=D_{3}+(D_{1}-D_{3})\frac{\partial C_{1,eq}}{\partial C_{1}}$$
(9)$$D_{1,2}=(D_{1}-D_{3})\frac{\partial C_{1,eq}}{\partial C_{2}}$$
(10)$$D_{2,1}=(D_{2}-D_{3})\frac{\partial C_{2,eq}}{\partial C_{1}}$$
(11)$$D_{2,2}=D_{3}+(D_{2}-D_{3})\frac{\partial C_{2,eq}}{\partial C_{2}}$$
where *C*_1_ and *C*_2_ are the stoichiometric concentrations of NaSMR and *β*-CD, respectively, and *D*_i_ (i = 1, 2 or 3) are the diffusivity of the species actually present in the solution.

Considering that the volume of the supramolecular complex (NaSMR:*β*-CD) is approximately equal to the sum of isolated NaSMR and CD molecule volumes, the diffusion coefficient of complexes (*D*_3_) can be estimated from the limiting binary values of NaSMR (*D*_1_ = 0.944 × 10^−9^ m^2^s^−1^) and *β*−CD (*D*_2_ = 0.326 × 10^−9^ m^2^s^−1^), by using Equation (12) [34]
(12)$$D_{3}={\left(D_{1}^{-3}+D_{2}^{-3}\right)}^{-1/3}$$

Table 5 shows the estimated values for the limiting diffusion coefficients of the free and complexed species.

A good agreement between these experimental and predicted values for parameters *D*_11_, *D*_12_, *D*_21_ and *D*_22_ using an equilibrium constant equal to 20 mol^−1^ dm^3^ is found (deviations 4%). As an illustrative example of this achievement, Figure 4 shows the experimental and predicted (using different values for *K*, i.e., 10, 20, 30, 50 and 100 mol^−1^ dm^3^) cross-diffusion coefficients *D*_12_ as a function of the solute fraction of NaSMR, *X*_1_ [33].

Despite the limitations of this theoretical framework (e.g., only valid in dilute solutions), this model is useful because it is possible from it to obtain an estimation of the association constant of these supramolecular structures, helping to understand its transport behaviour in aqueous solutions.

Pollen tube germination is a crucial step in the sexual reproduction of spermatophytes that has been widely studied [36,37]. This complex physiological process is influenced by several factors, namely calcium ions and boric acid [38,39]. As cyclodextrins can interact with several types of molecules [40], it is possible that the lower rate of germination observed on cyclodextrin-supplemented medium is due to the sequestration of calcium ions or boric acid and not due to toxicity effects. In fact, previous studies have shown the formation of non-covalent complexes of *β*-CD and divalent metal cations [41], and these by mass spectrometry and the overall structure of the complexes has been investigated by functional theory modelling, showing several possible structural conformations [42]. Future studies should explore this possible mechanism more deeply. On the other hand, sulphonamide antibiotics have been shown to impact folate pathways in *Arabidopsis thaliana* by epigenetic silencing mechanisms [43], which could explain the lower germination rates of pollen exposed to sulfamerazine. Taken together, the results of the pollen assays seem to show a lower toxicity of sulfamerazine–cyclodextrin complexes.

The results indicate that the weight of BY-2 cells is influenced by the different experimental conditions tested. Treatment with *β*-CD 0.009 mol dm^−3^ and HP-*β*-CD 0.01 mol dm^−3^ led to a decrease in cell weight, suggesting a potential inhibitory effect on cell growth. In contrast, treatment with *β*-CD 0.005 mol dm^−3^ resulted in an increase in cell weight, indicating a stimulatory effect. Previous studies have shown that CD can enhance the production of secondary metabolites, namely terpenes, in *Solanum lycopersicum* (tomato) culture cells, supporting these findings [44]. In fact, CD molecules have been used to encapsulate specific plant growth regulators, such as cytokinins [45]. The varying effects of sulfamerazine treatments on cell weight may be attributed to its complex interactions with the cells. The observed differences highlight the importance of considering the concentration and type of treatment when studying the growth dynamics of BY-2 cells.

## 5. Conclusions

The diffusion coefficients *D*_11_, *D*_12_, *D*_21_ and *D*_22_ were obtained for the aqueous ternary system NaSMR/CDs (i.e., *β*-CD and HP-*β*-CD). From these, it is evident that the diffusion behaviour of NaSMR in aqueous solutions is affected by the presence of the oligosaccharide *β*-CD. Comparing these results with those obtained for the binary systems at the same temperature using the same technique, it can be seen that the main coefficients *D*_11_ differ from the corresponding binary diffusion coefficients. In addition, coupled diffusion of NaSMR and *β*-CD occurs, as indicated by non-zero values of the cross-diffusion coefficients, *D*_12_. *β*-CD concentration gradients produce significant counter-current coupled flows of NaSMR. From this, we can conclude that *β*-CD influences the diffusion of the solution components and, under these circumstances, suggests that interaction between the solutes leads to a counterflow NaSMR. However, the diffusion data also show that there is no interaction between this drug and HP-*β*-CD, and the probability of the inclusion of NaSMR in the cavity of the sterically hindered HP-*β*-CD is very low. We believe that this provides transport data relevant to modelling the diffusion in pharmaceutical applications.

Combinations of sulfamerazine with *β*-CD (at certain concentrations) and HP-*β*-CD showed positive effects on kiwi pollen germination compared with isolated sulfamerazine.

The impact of *β*-CD appears variable depending on the concentration: while a combination of sulfamerazine 0.005 + *β*-CD 0.005 has positive effects on germination, the combination of sulfamerazine 0.01 + *β*-CD 0.009 mol dm^−3^ demonstrated a reduction in germination, indicating potential toxicity compared with isolated sulfamerazine. The effectiveness of these combinations may depend on the specific concentrations of sulfamerazine and cyclodextrins used, requiring careful adjustment of these concentrations to optimise the desired effects.

The results suggest that combinations of sulfamerazine with *β*-cyclodextrin have the potential to reduce sulfamerazine toxicity. However, additional analyses are necessary to confirm these observations and fully understand the effects of different combinations.

The weight of BY-2 cells is influenced by the experimental conditions, including the type and concentration of treatment. *β*-CD 0.009 dm^−3^ and HP-*β*-CD 0.01 mol dm^−3^ treatments resulted in decreased cell weight, while *β*-CD 0.005 mol dm^−3^ treatment led to an increase. Sulfamerazine treatments exhibited varied effects on cell weight. These findings contribute to our understanding of the factors affecting the growth dynamics of BY-2 cells and underscore the importance of considering treatment parameters in future research.

The results obtained from these studies on the interaction between sulfamerazine and cyclodextrins, along with the observed effects on BY-2 cells and pollen germination, have implications beyond the purely pharmaceutical scope. Considering that sulfamerazine is present in water and widely used in veterinary medicine, especially in animals such as horses, it is crucial to assess its environmental impact. Through understanding the effects of sulfamerazine on biological systems like plant cells and pollen germination, valuable insights can be gained into its potential impact on aquatic and terrestrial ecosystems. Furthermore, the results highlight the relevance of tests on plant cells and pollen germination as important tools in evaluating the environmental effects of pharmaceutical compounds, enabling a more holistic approach to managing and mitigating risks associated with the use of these products.

## Figures and Tables

**Figure 1 biomolecules-14-00462-f001:**
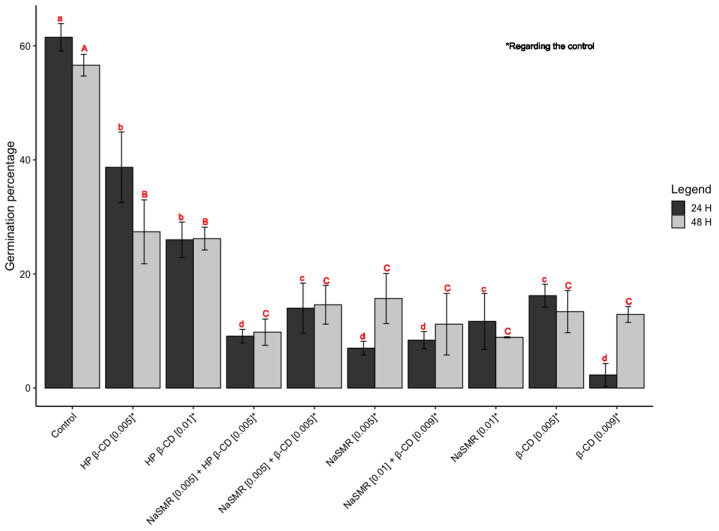
Results, in percentages, of pollen grains germinated under various experimental conditions. Different letters (a–d and A–C) for the same time period indicate statistically different results by Tukey’s test (*p* < 0.05).

**Figure 2 biomolecules-14-00462-f002:**
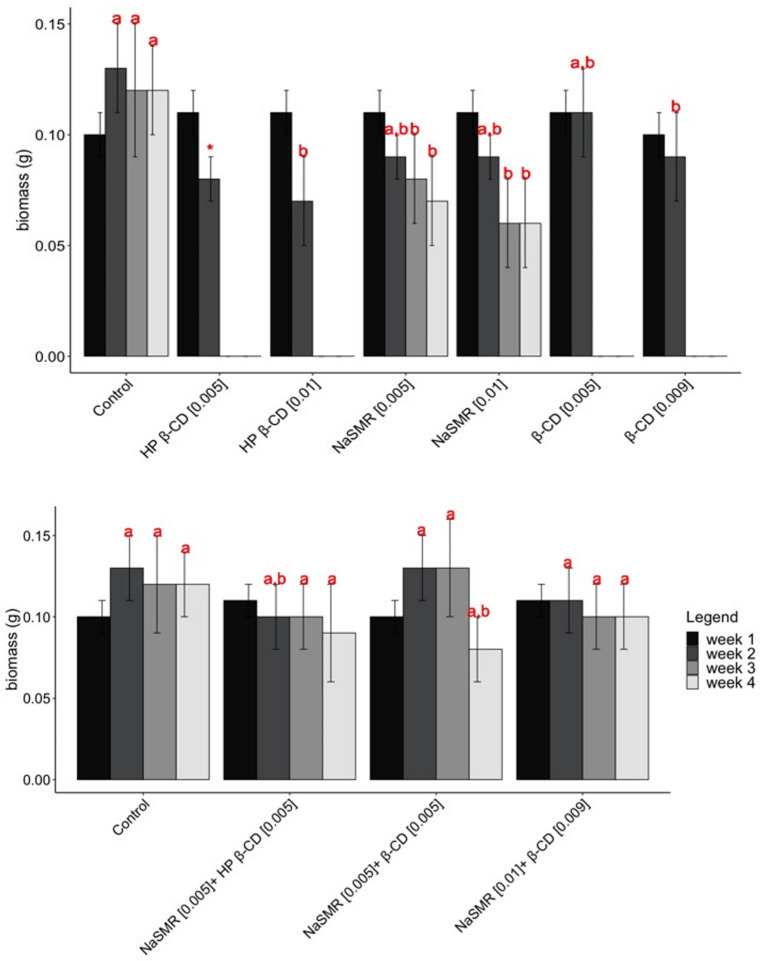
BY-2 cell mass (g) when exposed to different experimental conditions. Different letters (a and b) for the same time period indicate statistically different results by Tukey’s test (*p* < 0.05).

**Figure 3 biomolecules-14-00462-f003:**
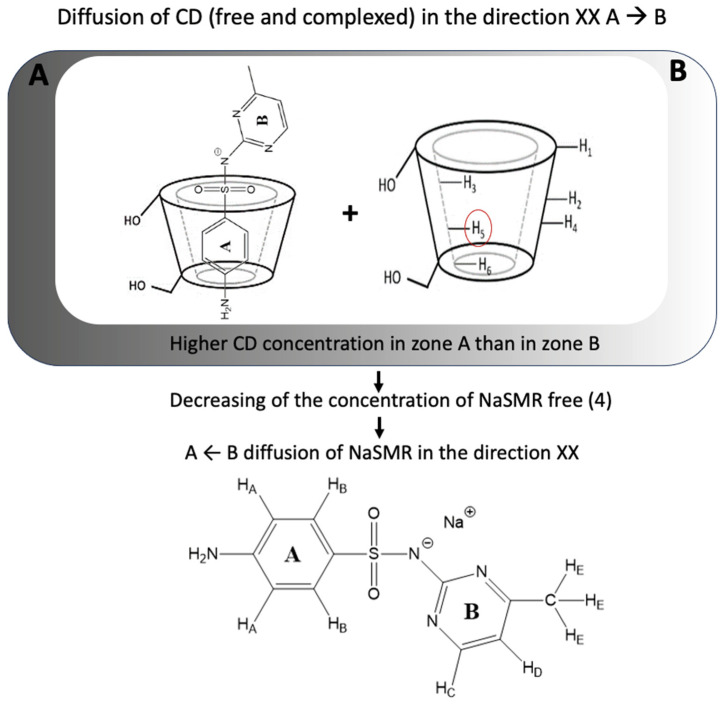
Schematic representation of the counterflow of NaSMR in solutions containing cyclodextrins (CDs).

**Figure 4 biomolecules-14-00462-f004:**
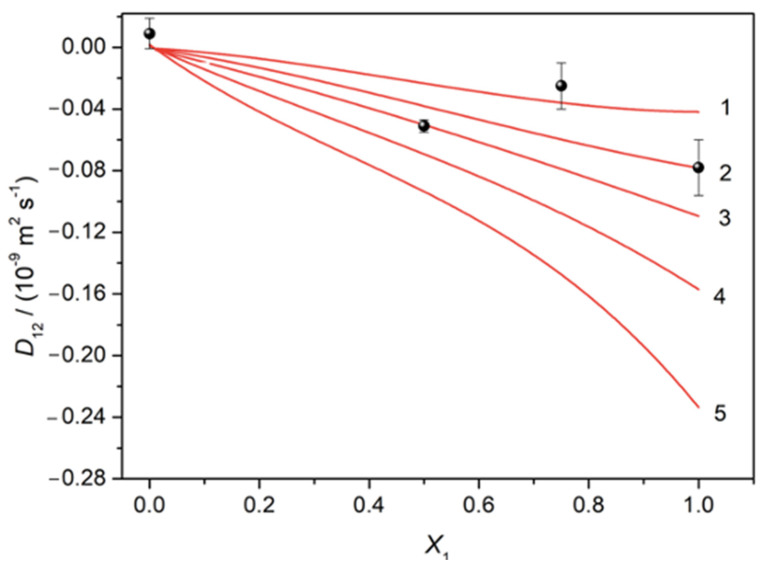
Effect of different binding constants on the prediction of cross-diffusion coefficients (*D*12) as a function of *X*_1_ and comparison with experimental data (circles). Lines from top to bottom: *K* = 10 (1), 20 (2), 30 (3), 50 (4) and 100 (5) dm^3^ mol^−1^.

**Table 1 biomolecules-14-00462-t001:** Sample description.

Chemical Name	Source	CAS Number	Mass Fraction Purity ^1^
Sulfamerazine sodium salt	Sigma-Aldrich	127-58-2	0.99
*β*-Cyclodextrin	Sigma-Aldrich (water mass fraction 0.131) ^2^	7585-39-9	>0.97
HP-*β*-Cyclodextrin	Sigma-Aldrich (water mass fraction 0.03) ^3^	128446-35-5	>0.97
Water	Millipore-Q water (18.2 mΩ·cm at 25.00 °C)		
Potassium nitrate	Sigma-Aldrich	7757-79-1	0.99
Magnesium sulphate heptahydrate	Sigma-Aldrich	10034-99-8	0.98
Calcium nitrate tetrahydrate	Merck	13477-34-4	0.99
Boric acid	Merck	10043-35-3	0.995
Sucrose	Duchefa Biochemie B.V	57-50-1	0.98
Plant agar	Duchefa Biochemie B.V	9002-18-0	

^1^ As stated by supplier. ^2^ The mass fraction purity is on water-free basis; these data are provided by the suppliers. ^3^ The water content was determined by Karl Fischer method in our laboratory, and the corresponding value obtained was taken into account to determine the solution concentration.

**Table 2 biomolecules-14-00462-t002:** Diffusion coefficients of sulfamerazine sodium salt in aqueous solutions of concentration C, at 25.00 °C and at pressure *P* = 101.3 kPa.

*C/*(mol dm^−3^)	*D* ± SD/(10^−9^ m^2^s^−1^) ^1^
0.000	0.944 ± 0.003
0.005	0.927 ± 0.005
0.008	0.917 ± 0.002
0.010	0.906 ± 0.003

^1^ *D* is the mean diffusion coefficient value of sulfamerazine in aqueous solutions using Millipore water from three experiments, and SD is the standard deviation of that mean.

**Table 3 biomolecules-14-00462-t003:** Ternary mutual diffusion coefficients *D*_11_, *D*_12_, *D*_21_ and *D*_22_ for aqueous NaSMR + *β*-CD or HP-*β*-CD (*C*_2_) solutions and the respective standard deviation of the mean SD at 25.00 °C and at pressure *P* = 101.3 kPa.

*C*_1_ ^a^	*C*_2_ ^a^	*X*_1_ ^b^	*D*_11_ ± SD ^c^	*D*_12_ ± SD ^c^	*D*_21_ ± SD ^c^	*D*_22_ ± SD ^c^
NaSMR (component 1) + *β*-CD (component 2)						
0.000	0.010	0.000	0.878 ± 0.004	0.010 ± 0.010	0.022 ± 0.015	0.330 ± 0.004
0.005	0.005	0.500	0.888 ± 0.004	0.011 ± 0.002	0.024 ± 0.014	0.335 ± 0.006
0.0075	0.0025	0.750	0.892 ± 0.002	−0.050 ± 0.003	0.030 ± 0.013	0.339 ± 0.001
0.010	0.000	1.000	0.946 ± 0.005	−0.080 ± 0.002	−0.008 ± 0.003	0.338 ± 0.012
NaSMR (component 1) + HP-*β*-CD (component 2)						
0.000	0.010	0.000	0.864 ± 0.008	0.021 ± 0.028	0.008 ± 0.001	0.325 ± 0.009
0.005	0.005	0.500	0.909 ± 0.006	0.026 ± 0.030	0.010 ± 0.015	0.323 ± 0.004
0.0075	0.0025	0.750	0.920 ± 0.006	0.015 ± 0.030	0.016 ± 0.011	0.320 ± 0.003
0.010	0.000	1.000	0.930 ± 0.016	0.020 ± 0.010	0.002 ± 0.004	0.327 ± 0.015

^a^ *C*_i_ in units of mol dm^−3^. ^b^ *X*_1_ = *C*_1_/(*C*_1_ + *C*_2_) represents the NaSMR solute mole fraction. ^c^ Mean diffusion coefficients from four to six replicate measurements in units of 10^−9^ m^2^s^−1^. Standard uncertainties u are u_r_(*C*) = 0.03; u(*T*) = 0.01 K and u(*P*) = 2.03 kPa.

**Table 4 biomolecules-14-00462-t004:** Estimation of the number of moles of each component transported per mole of the other component in aqueous ternary systems NaSMR (*C*_1_) + *β*-CD (or HP-*β*-CD) (*C*_2_).

*C*_1_ ^a^	*C*_2_ ^a^	*D*_12/_*D*_22_ ^b^	*D*_21/_*D*_11_ ^c^
NaSMR (component 1) + *β*-CD (component 2)			
0.0000	0.0100	0.030	0.025
0.0050	0.0050	0.032	0.027
0.0100	0.0090	−0.148	0.034
0.0100	0.0000	−0.237	−0.010
NaSMR (component 1) + HP-*β*-CD (component 2)			
0.000	0.010	0.065	0.009
0.005	0.005	0.080	0.011
0.0075	0.0025	0.046	0.012
0.010	0.000	0.061	0.002

^a^ *C*_i_ in units of mol dm^−3^. ^b^ The number of moles of component 1 (NaSMR) transported per mole of component 2 (*β*-CD or HP-*β*-CD). ^c^ The number of moles of component 2 (*β*-CD or HP-*β*-CD) transported per mole of NaSMR.

**Table 5 biomolecules-14-00462-t005:** Limiting diffusion coefficients, *D*_i_ (i = 1, 2 or 3), of species at 25.00 °C.

Species	*D*_i_/(10^−9^ m^2^s^−1^)
NaSMR	0.944
*β*-CD	0.326 [35]
NaSMR:*β*-CD	0.322

## Data Availability

The data presented in this study are available on request from the corresponding author.

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
