# Peer review of "On Interactions of Sulfamerazine with Cyclodextrins from Coupled Diffusometry and Toxicity Tests"

_biomolecules, 2024, doi:10.3390/biom14040462_

Round 1

Reviewer 1 Report

Comments and Suggestions for Authors

In the paper entitled “On interactions of sulfamerazine with cyclodextrins from coupled diffusometry and toxicity tests: its impact on biological systems”, the diffusion behavior of sulfamerazine in aqueous solutions of two cyclodextrins was studied. The authors revealed a positive effect of complexed sulfamerazine on the germination of kiwi pollen compared to pure sulfamerazine. In addition, the effect of the complexes on the weight of BY-2 cells was studied. The manuscript needs serious revision because the following major and minor questions and comments have arisen, listed below.

Major:

1. Is it possible to tell us in more detail about counter-flow of sulfamerazine and present this process schematically in the form of figure?

2. Why do the authors call the HP-β-CD cavity sterically hindered? What prevents the inclusion of the relatively small molecule sulfamerazine? HP-β-CD generally encapsulates guests better than β-CD.

3. For the phrase “Cyclodextrins, namely beta-cyclodextrin (β-CD) and hydroxypropyl beta-cyclodextrin (HP-β-CD), have been used for this purpose,” the literary references confirming the biosafety of β-CD and HP-β-CD are required.

4. If there is published data about sulfamerazine complexes with cyclodextrins, then it is worth writing about these complexes in the Introduction section.

5. Section 2 lacks a description of the preparation of the complexes.

6. What molecular weights of β-CD and HP-β-CD were taken to calculate their moles?

7. Are there any literature data with the self-diffusion coefficients of sulfamerazine and cyclodextrins obtained using NMR and dynamic light scattering? If so, it would be interesting to look at a comparison of self-diffusion coefficient values obtained by different methods.

8. Is it possible to suggest a mechanism for the sequestration of calcium or boric acid ions in this manuscript?

9. Conclusions should be concise and understandable. If the title of the manuscript contains “its impact on biological systems,” then the authors need to write more clearly about this impact on biological systems.

Minor:

1. Line 39. forming a hydrophobic

2. Line 286. Figure 3.

Author Response

Reply to Reviewers’ comments

 Reviewer #1 On the paper entitled “On interactions of sulfamerazine with cyclodextrins from coupled diffusometry and toxicity tests: its impact on biological systems”, the diffusion behavior of sulfamerazine in aqueous solutions of two cyclodextrins was studied. The authors revealed a positive effect of complexed sulfamerazine on the germination of kiwi pollen compared to pure sulfamerazine. In addition, the effect of the complexes on the weight of BY-2 cells was studied. The manuscript needs serious revision because the following major and minor questions and comments have arisen, listed below.

We are grateful for these positive comments.

Major:

Is it possible to tell us in more detail about counter-flow of sulfamerazine and present this process schematically in the form of figure?

We agree with the referee and, consequently, we have put together one scheme to show this process (Figure 3). We can say that negative values for D12 may be understood by considering transport along cyclodextrin gradients formed in solutions of NaSMR. In the region of the solution with the higher cyclodextrin concentration, most of the free SMR is consumed by formation of SMR:CD supramolecular complexes. As a result, the gradient in cyclodextrin produces a gradient with opposite sign in the concentration of NaSMR species. Hence D12 values are negative for these solutions.

  1. Why do the authors call the HP-β-CD cavity sterically hindered? What prevents the inclusion of the relatively small molecule sulfamerazine? HP-β-CD generally encapsulates guests better than β-CD.

We are grateful for these positive comments. In fact, the referee is right, concerning to this question about the inclusion of the relatively small molecule sulfamerazine in HP-β-CD cavity. However, from Dij almost zero data, we can hypothesize that for the considered concentration range, there are no complex formation.

A possible explanation for obtaining Dij almost zero data than would be expected (when compared to the others), can be attributed to the fact that it was only used in this work lower concentrations of the diffusing species. For this range of concentrations, these results show us that the HP-β-CD concentration gradients cannot drive significant coupled flows of NASMR and, consequently, leading to unfavourable conditions to formation of inclusion complexes with this sterically hindered cyclodextrin in solution. The effect HP-β-CD on the motion of this drug may be associated with the obstruction that these large molecules exert on the motion of the small one.

Support for this comes from other molecules and other techniques. For example, for in system containing L-dopa and HP-β-CD, the behaviour of diffusion is in complete agreement with the 1H NMR spectroscopy results, which suggest that there is, effectively, no complexation between HP-β-CD and L-dopa. (Ldopa) https://doi.org/10.1016/j.jct.2016.01.010

  1. For the phrase “Cyclodextrins, namely beta-cyclodextrin (β-CD) and hydroxypropyl beta-cyclodextrin (HP-β-CD), have been used for this purpose,” the literary references confirming the biosafety of β-CD and HP-β-CD are required.

We agree with the referee and, consequently, the literary references have been inserted accordingly.

  1. If there is published data about sulfamerazine complexes with cyclodextrins, then it is worth writing about these complexes in the Introduction section.

 We agree with the referee and, consequently, the text has been modified accordingly.

  1. Section 2 lacks a description of the preparation of the complexes.

The authors do not understand this reviewer's comments and are therefore quite puzzled by them. In fact, in the present work, we did not synthetise complexes in a solid state by different methods.  Despite possibly be interesting, however, is not part of the aim of this work.

  1. What molecular weights of β-CD and HP-β-CD were taken to calculate their moles?

The referee is right and, consequently, we have inserted the weights of β-CD and HP-β-CD in experimental section.

  1. Are there any literature data with the self-diffusion coefficients of sulfamerazine and cyclodextrins obtained using NMR and dynamic light scattering? If so, it would be interesting to look at a comparison of self-diffusion coefficient values obtained by different methods.

We are grateful for these comments and allow us to clarify this point.

It is very common to find misunderstandings concerning the meaning of a parameter, frequently just denoted by D and merely called diffusion coefficient, in the scientific literature, communications, meetings, or simple discussions among researchers. In fact, it is necessary to distinguish self-diffusion D* (intradiffusion, tracer diffusion, single ion diffusion, ionic diffusion) and mutual diffusion D (interdiffusion, concentration diffusion, salt diffusion). Many techniques are used to study diffusion in aqueous solutions. Methods such as NMR, polarographic, and capillary-tube techniques with radioactive isotopes measure self-diffusion coefficients (“intradiffusion coefficients”). However, for bulk ion transport the appropriate parameter is the mutual diffusion coefficient, D. Relationships derived between intradiffusion and mutual diffusion coefficients, D* and D, have had limited success and consequently mutual diffusion coefficients are much needed.

Experimental methods that can be employed to determine mutual diffusion coefficients are: Diaphragm-Cell (inaccuracy 0.5-1%), Conductimetric (inaccuracy 0.2%), Gouy and Rayleigh Interferometry (inaccuracy <0.1 %) and Taylor Dispersion (inaccuracy 1-3%). While the first and second methods consume days in experimental time, the last ones imply just hours. The conductimetric technique follows the diffusion process by measuring the ratio of electrical resistances of the electrolyte solution in two vertically opposed capillaries as time proceeds. Despite this method has previously given us reasonably precise and accurate results, it is limited to studies of mutual diffusion in electrolyte solutions, and like diphragm-cell experiments, the run times are inconveniently long (~days). The Gouy Method also has high precision, but when applied to microemulsions they are prone to gravitational instabilities and convection. Thus, the Taylor dispersion has become increasingly popular for measuring diffusion in solutions, because of its experimental short time and its major application to the different systems (electrolytes or non-electrolytes). In addition, with this method it is possible to measure multicomponent mutual diffusion coefficients.

In conclusion, the Taylor dispersion is a rapid, inexpensive and convenient technique for multicomponent diffusion measurements. In the present work we have measured mutual diffusion coefficients by using this technique.

Regarding the clarification needed to the choosing the Taylor dispersion method over alternative techniques for studying diffusion, some references and figures were included in the text. It is important to note that self-diffusion coefficient values obtained by different methods (e.g., NMR) and mutual diffusion from Taylor have different meanings. Going back to the question, it can be found in the literature self-diffusion or intra diffusion coefficients for cyclodextrins but, at the best of our knowledge, not for sulfamerazine.

  1. Is it possible to suggest a mechanism for the sequestration of calcium or boric acid ions in this manuscript?

We are grateful for these comment, sequestration of bivalent metals has been described by non-covalent biding in the literature. A small discussion and appropriate sources have been added.

  1. Conclusions should be concise and understandable. If the title of the manuscript contains “its impact on biological systems,” then the authors need to write more clearly about this impact on biological systems.

We agree with the referee and the title has been changed and additional information has been added to the conclusions.

Minor:

  1. Line 39. forming a hydrophobic
  2. Line 286. Figure 3.

We do apologize for the typos. We did our best to correct all of them throughout the ms.

Reviewer 2 Report

Comments and Suggestions for Authors

The present work is on a topic of interest, but it lacks studies to justify its results (I have seen low discussion with available data in bibliography for example) and a better analysis of its own results, to merit being published in biomolecules:

In general, the methods should be better detailed. Details are missing that can ensure its reproducibility.

1. No evidence is presented for the formation of the CD:Sulfamerazine inclusion complex, although the authors talk about it by adding what appears to be a co-formulation. I cannot see also the calculated K.

2. A statistical study of the Taylor analysis and toxicity is necessary to better understand the results; it is difficult to interpret differences since they are all so close to each other.

3. The diffusion study is difficult to understand explained, it would be a good idea to add additional graphic material to see the steps better.

4. DCs' toxicity needs to be better justified and discussed. Do cyclodextrins remove calcium from the environment?

5. There are mistakes in the figure ordination.

6. Figure "2", please use the standard units for K.

7. The conclusions must be consistent with the results. Do adding bCD and sulfamerazine have positive effects on germination? The results seem to indicate the opposite. Especially because, although there is no statistical analysis, after 48 hours it seems that they have all germinated the same...

Finally, some references do not follow the journal's standard.

As said, the work needs a deeper discussion of its results, re-evaluating them and adjusting to what they see to be considered in this journal.

Author Response

Reply to Reviewers’ comments

 Reviewer #2

 Comments and Suggestions for Authors

The present work is on a topic of interest, but it lacks studies to justify its results (I have seen low discussion with available data in bibliography for example) and a better analysis of its own results, to merit being published in biomolecules: In general, the methods should be better detailed. Details are missing that can ensure its reproducibility.

1.No evidence is presented for the formation of the CD:Sulfamerazine inclusion complex, although the authors talk about it by adding what appears to be a co-formulation. I cannot see also the calculated K.

We are grateful for these comments. First, we would appreciate to introduce some notes having in mind to clarify some points. In true, relative to estimation of K, we can consider that is a trial-and-error method of nature semi-empirical, where simple expressions correlate the experimental D12 with association constant K for the equilibrium between NaSMS and CDs. In the other words, the four-diffusion coefficients measured were used to estimate values of the binding constant, K. For more details see 10.1002/bbpc.19900940706. The best value found for K was 20 mol-1 dm3.

For particular aqueous system containing NASMS plus β-CD, from the values K not zero, D12 values negative can be interpreted by the formation of aggregates between this drug and β-CD molecules. In conclusion, we can say from some experimental evidence, that the complexation is a phenomenon which can be responsible for D12 negative values.

Support for the evidence experimental and theoretical that there is inclusion complexation behaviour and binding ability of sulfamerazine (with α- and β-cyclodextrins) comes the literature (e.g., https://doi.org/10.1016/j.saa.2014.01.057).

However, relative the nature of these complexes, accordingly with the real objects of research, the referee is right and, consequently, we have modified the name, delete the word “inclusion”.

That is, supramolecular complexes instead inclusion complexes.

  1. A statistical study of the Taylor analysis and toxicity is necessary to better understand the results; it is difficult to interpret differences since they are all so close to each other.

Thanks for the comment.

First, we do apologize for the mistake in the Table 2. That is, it must read the value D12 = −(0.080±0.002) × 10−9 m2 s−1   instead of D12 =−(0.080±0.025) × 10−9 m2 s−1.

Relative to the statistical study of the Taylor analysis proposed by the referee (which would be interesting but it is outside of this scope of paper), we consider that the information given in Table 2 is already sufficient to help to interpret these differences. Table 2 summarizes the diffusion coefficient values and the respective standard deviations of the means, D11, D12, D21, D22 for the ternary NaSMR+ CDs + water, at 298.15 K. These results are, in general, the average ones from 4 independent experiments. Main diffusion coefficients D11 and D22 were generally reproducible within (± 0.010 × 10−9 m2 s−1). Cross-diffusion coefficients D12 and D21, describing the coupled diffusion of drug and H-β-CD, were reproducible within about (± 0.030×10−9 m2 s−1). These last standard deviation values are close to D12 and D21 values, and hence, we can consider that are almost zero, within the experimental error. However, for -β-CD /NaSMR, the standard deviations for D12 are lower (e.g., at X1= 1.0, D12 =−0.050±0.003×10−9 m2 s−1), permitting to say that in this particular case, D12 is not zero.

In addition, we would appreciate introducing some notes, having in mind to clarify yet this question. Binary aqueous solution of KCl 0.100 mol dm−3 was used to test the operation of the dispersion equipment. This system was chosen because its diffusion coefficient is accurately known from conductimetric experiments (uncertainties <0.5%) Comparison of the results (shown in Material Supplementary) suggests an acceptable uncertainty of 1–3% for the Taylor D values for binary systems, having in mind that 1–3% uncertainty is typical for Taylor dispersion measurements.

Thus, despite of tests of the optimization on dispersion equipment will be usually done before starting a new system, and thus, we did not indicate these results, we have now inserted this additional information in Appendix B.

The general tendency is shown, for example high concentration of HB-β-CD completely

inhibit BY-2 growth. Although the statistical analyses can be made, the main objective of

these assays were to access if the antibiotic alone or in CD mixtures has in impact on

biological systems. These relations should be investigated further and in more molecular

detail in other works.

  1. The diffusion study is difficult to understand explained, it would be a good idea to add additional graphic material to see the steps better.

The referee is right and, consequently, we have now inserted this additional information in Appendix A.

  1. DCs' toxicity needs to be better justified and discussed. Do cyclodextrins remove calcium from the environment?

Thanks for the comment. Although the sequestration has not been directly observed in this work, non-covalent biding between CDs and metal ions has been described in the literature. As stated, it is a possible mechanism that should be furthered investigated.

  1. There are mistakes in the figure ordination.

We do apologize for the typos. We did our best to correct all of them throughout the ms.

  1. Figure "2", please use the standard units for K. M-1 or mol-1 dm3

We understand the comments raised by the Reviewer. However, the measurements were done at different activity coefficients; consequently, the best way to present these binding (or association) constants is by adding dimensions.

  1. The conclusions must be consistent with the results. Do adding b-CD and sulfamerazine have positive effects on germination? The results seem to indicate the opposite. Especially because, although there is no statistical analysis, after 48 hours it seems that they have all germinated the same...

Thanks for the comment. The results are presented as percentage of germination in terms of the control. All molecules, b-CD and sulfamerazine included, have a negative effect on the germination. HP-b-CD shows lesser inhibition and there appears to be a less negative impact than on sulfamerazine alone. (Statistical analysis could be added, but it appears to be tangential to the main discussion).

  1. Finally, some references do not follow the journal's standard.

We do apologize for the typos. We did our best to correct all of them throughout the ms.

As said, the work needs a deeper discussion of its results, re-evaluating them and adjusting to what they see to be considered in this journal.

We appreciate all your comments. Despite the study of the type of interactions as well as the morphological characteristics of the complexes of cyclodextrin with different drugs, including NaSMS, is outside of this scope of the paper, we can say that this matter is well documented in the literature. For example, see the references indicated in the present work.

However, contrary to this situation, as far the authors know, there are no data of mutual diffusion. Thus, these measurements are justified considering that this information is very important for the irreversible thermodynamic behaviour of the involved species, not so much the complex question of the nature of their internal binding forces, and, of course to the knowledge of the structure of these solutions.

Relative to the reasons for choosing the two cyclodextrins, these compounds were used due to the possibility of interacting with this drug, thus allowing their capture in the medium. Despite being a very simple model, we believe that with these data we can contribute to a better understanding of these systems and, hence, the possibility of reducing toxicity where different materials are used to combine with this drug. It should be stressed that the K values obtained in this way are very similar to those obtained by 1H NMR, as it was reported by us in several papers (e.g., DOI: 10.1016/j.jct.2015.06.022)

Round 2

Reviewer 1 Report

Comments and Suggestions for Authors

I am not satisfied with the responses to my comments 4 and 5.
1) If no one has previously studied complexes of sulfamerazine derivatives with any cyclodextrins, then this should be reported in the Introduction section.
2) The objective of this work is to investigate the behavior of sulfamerazine and cyclodextrins in aqueous solution. In the section Materials and Methods I would like to see a description of the preparation of these solutions. Was a solution of one substance mixed with a solution or with a solid sample of another substance?

Author Response

Reviewer #1

1) If no one has previously studied complexes of sulfamerazine derivatives with any cyclodextrins, then this should be reported in the Introduction section.

The referee is right when he says that if there is published data about sulfamerazine complexes with cyclodextrins, then it is worth writing about these complexes in the Introduction section.

In fact, there was a concern on our part to reference some works whose aim was to study these complexes (e.g., reference 3). However, having in mind more clarity, we have  inserted the following sentence in the Introduction.

“Although there are already some studies involving these compounds (e.g., [3, 17, 21]), to the best of our knowledge, no data exists on the ternary mutual diffusion coefficients of drug in aqueous solutions containing cyclodextrins. This present study aims to fill this gap by providing experimental data on the diffusion coefficients measured by the Taylor dispersion method for two ternary systems (NaSMR /b-CD/H2O and NaSMR /HP-b-CD/H2O) at carrier concentrations of 0.000 mol dm-3 and 0.010 mol dm-3 at 25.00 ºC.

2) The objective of this work is to investigate the behavior of sulfamerazine and cyclodextrins in aqueous solution. In the section Materials and Methods I would like to see a description of the preparation of these solutions. Was a solution of one substance mixed with a solution or with a solid sample of another substance?

The referee is right; consequently, we have inserted a summary description of the preparation of these solutions in the manuscript. That is:

“Binary (NaSMR /H2O and KCl /H2O), and ternary solutions ((NaSMR /b-CD/H2O and NaSMR /HP-b-CD/H2O) were prepared by weighing the appropriate amounts of the solutes and after dissolving in water to finally obtain the desired molar concentration”.

Reviewer 2 Report

Comments and Suggestions for Authors

Thanks to the authors for the comments.

Only the statistics part remains, no matter how much the authors say that it is outside the scope of the work, the correct statistical treatment of all the data is necessary to ensure a correct interpretation of the data and not draw false conclusions.

I actively suggest the correct statistical preparation of the work before publishing it.

Author Response

One way analysis of variance (ANOVA) has been carried out in pollen germination and BY-2 growth assays. Although the high standard deviation of the biological samples and the reduced sample number may add some uncertainty, the general trend of the results shows that both CD and antibiotic are growth inhibitors and in the case of BY-2, CDs are particularly relevant, likely due to the ion sequestration or similar effects that are already discussed. Nonetheless, when combined with sulfamerazine, there are observable growth benefits.